



# Uncertainty analysis of floodplain friction in hydrodynamic models

Guilherme Dalledonne[1], Rebekka Kopmann[1], and Thomas Brudy-Zippelius[1]

[1]Department of Hydraulic Engineering in Inland Areas, Federal Waterways Engineering and Research Institute, BAW, Karlsruhe, Germany

**Correspondence:** G. Dalledonne (guilherme.dalledonne@baw.de)

**Abstract.** This study proposes a framework to estimate the uncertainty of hydrodynamic models on floodplains. The traditional floodplain resistance formula of Pasche (1984) (based on Lindner, 1982) used for river modelling as well as the approaches of Baptist et al. (2007), Järvelä (2004), and Battiato and Rubol (2014) have been considered for carrying out an uncertainty analysis (UA). The analysis was performed by means of three different methods: traditional Monte Carlo (MC), First-Order Second-Moment (FOSM) and Metamodelling. Using a two-dimensional hydrodynamic model, a 10 km reach of the River Rhine was simulated. The model was calibrated with water level measurements under steady flow conditions and then the analysis was carried out based on flow velocity results. The compared floodplain friction formulae produced qualitatively similar results, in which uncertainties in flow velocity were most significant on the floodplains. Among the tested resistance formulae the approach from Järvelä presented on average the smallest prediction intervals i.e. the most accurate results. It is important to keep in mind that UA results are not only dependent on the defined input parameters deviations, but also on the number of parameters considered in the analysis. In that sense, the approach from Battiato and Rubol is still attractive for it reduces the current analysis to a single parameter, the canopy permeability. The three UA methods compared gave similar results, which means that FOSM is the less expensive one. Nevertheless it should be used with caution as it is a first-order method (linear approximation). In studies involving dominant non-linear processes, one is advised to carry out further comparisons.

## 1 Introduction

Flow resistance can be considered as the contribution of four components, according to Rouse (1965): (a) surface, (b) form, (c) wave, and (d) flow unsteadiness resistances. Not only that, it is a complex phenomenon dependent on Reynolds number, relative roughness, cross-sectional geometry, channel non-uniformity, Froude number, and degree of flow unsteadiness. Also, Yen (2002) affirms that flow resistance interacts "in a non-linear manner such that any linear separation and combination is artificial".

There are several approaches available in the literature for determining flow resistance coefficients of vegetated floodplains in numerical models. These approaches are basically divided under four categories: rigid or flexible, and emergent or submerged vegetation. They aim to determine the resistance exerted by the vegetation on the flow based on physical properties such as vegetation height and width, stem diameter and density, etc. Recent research on flow resistance of emergent floodplain





vegetation is given in Aberle and Järvelä (2013) and a review of vegetated flow models can be found in Nikora et al. (2007). For an overview of the main vegetation friction laws available the reader is referred to the review given in Shields et al. (2017).

Even though much work has been done in applying different approaches to include vegetation induced resistance effects in hydrodynamic calculations, the majority of these studies have been verified only under laboratory scale conditions. A gap between those results and river engineering projects still exists. While free surface information on flooded areas can be well approximated from river channel measurements, flow velocity cannot. And because floodplain measurements usually are not available, model performance is neglected at those areas. That means, when flood scenarios belong to the scope of a project or study, attention should be given to this matter. A way to address this problem is to consider a probabilistic approach and to carry out an uncertainty analysis (UA) of the floodplain friction. Uncertainty in the context of fluid dynamics is defined as a potential deficiency of the simulation process, according to Walters and Huyse (2002). Straatsma and Huthoff (2010) considered floodplain friction parametrization to be an important source of uncertainty. Also, Di Baldassarre et al. (2010) compared and discussed deterministic and probabilistic approaches for floodplain mapping. They concluded that due to uncertainties related to flood-event statistics the probabilistic approach was considered to be a more correct representation.

Some studies can be found in the literature involving uncertainty analysis related to floodplains and the resistance coefficient. Apel et al. (2004) presented a flood risk assessment by means of a simple hydrological flood routing model in the Lower Rhine applying a Monte Carlo (MC) framework. Pappenberger et al. (2005) conducted an uncertainty analysis using a one-dimensional hydraulic model using a generalised likelihood uncertainty estimation. Their results showed that many parameter sets (channel and floodplain) can perform equally well even with extreme values. Brandimarte and Di Baldassare (2012) showed that the deterministic approach underestimates the design flood profile in hydraulic modelling and proposed an alternative approach based on the use of uncertain flood profiles. Altarejos-García et al. (2012) used the Point-Estimate Method for carrying out uncertainty analysis as an alternative to MC approaches to get estimates of the mean and variance of water depth and velocity. They considered the roughness coefficient as the main source when assessing the uncertainty in river flood modelling. Domeneghetti et al. (2013) proposed a methodology to derive probabilistic flood maps taking into account several sources of uncertainty. Willis et al. (2016) concluded that hydrodynamic modelling can be improved by increasing the number of frictional surfaces; however, they draw attention to the numerical scheme choice, which might lead to much larger errors.

In this context, a framework to estimate the uncertainty of hydrodynamic models on floodplains due to vegetation is proposed in the current study. A two-dimensional hydrodynamic model is calibrated with floodplain friction formulations, to which uncertainties are associated. After defining variations for sensitive input parameters, the uncertainty analysis is carried out with different methods for comparison. In the next section four chosen floodplain resistance formulae are described and analysed. Then the concept of uncertainty analysis is briefly explained and three different methods are presented in the third part. The fourth section provides information on the case study including a brief description of the hydrodynamic model, parameters used for model calibration, and a definition of input parameter uncertainties needed for carrying out the analysis. In the fifth section results are presented and discussed, from which conclusions are drawn in the last part of the manuscript.





## 2 Floodplain friction

Vegetation found on river banks and floodplains plays an important role on flow velocity profile and, therefore, on hydraulic roughness. Current research aims to relate vegetated floodplain properties to their *hydraulic signatures* and to incorporate the complex nature of vegetation characteristics into floodplain friction models. According to Shields et al. (2017), there are

5 no established practices for defining flow-dependent vegetation roughness values and incorporating them into hydrodynamic models. Additionally, model calibration usually is carried out with measurements taken in the main channel, and seldom (if ever) on floodplains. Thus, model response on floodplains cannot be verified and only relative conclusions can be made. It is under these circumstances that UA is especially useful for quantifying the probability of results. Basically the available approaches for vegetation friction formulation are subdivided in emergent/submerged and rigid/flexible.

10 For the current study four floodplain friction formulations are considered: Lindner (1982) and Pasche (1984), Baptist et al. (2007), Järvelä (2004), and Battiato and Rubol (2014). The first approach is a recommended practice by the German Association for Water, Wastewater and Waste (DVWK, 1991) for hydraulic calculations and it is commonly used in the BAW's projects. The second and third approaches represent the rigid (Baptist et al.) and flexible (Järvelä) approximations. Lastly, the approach from Battiato and Rubol is chosen for it proposes a completely different concept based on porous medium flow.

### 15 2.1 Lindner and Pasche

The modified formulation from Pasche (1984), based on Lindner (1982), was developed for rigid emergent vegetation. The Darcy-Weisbach friction factor for vegetation ($f_v$) can be obtained after the bulk drag coefficient ($C_D$) is iteratively calculated by the following equations:

$$U = \sqrt{\frac{8gR_hS_0}{f}}, \quad \text{with} \quad f = f_b + f_v$$

$$\left(\frac{U_i}{U}\right)^2 = 1.15\left(W_l\sqrt{m}\right)^{-0.48} + 0.5\left(W_w\sqrt{m}\right)^{1.1}$$

$$0.03 = 0.9\left(\frac{W_l}{C_{D1}}\right)^{-0.7}\left(1 + \frac{2gW_lS_0}{U^2}\right)^{-1.5}$$

$$W_w = 0.24W_l^{0.59}(C_{D1}D)^{0.41}$$

$$\text{Fr} = \frac{U}{\sqrt{gh}} \simeq \frac{y^*(y^{*2}-1)}{2[y^* - 1/(1 - D\sqrt{m})]}$$

$$C_D = 1.31\left(\frac{U_i}{U}\right)^2 + 2\text{Fr}^{-2}(1 - y^*)$$

25
$$f_v = 4C_DmDh \tag{1}$$

where $R_h$ is the hydraulic radius, $S_0$ is the bottom slope, $f_b$ is the bottom friction, $U$ is the approach velocity (upstream), $U_i$ is the calculated velocity (downstream), $W_l, W_w$ are the wake length and width resp., $C_{D1}$ is the drag coefficient for a single stem, $m$ is the number of stems per $\text{m}^2$, $D$ is the stem diameter, $h$ is the water depth and $g$ is the gravitational acceleration.





## 2.2  Baptist et al.

The approach from Baptist et al. (2007) was developed for rigid vegetation. They modelled the vegetation resistance force as the drag force on an array (random or staggered) of rigid cylinders with uniform properties. The velocity profile is calculated for two conditions: non-submerged (emergent) and submerged vegetation. For the case of emergent vegetation a uniform velocity

is assumed. For the case of submerged vegetation the velocity profile is subdivided in a uniform velocity zone (within the vegetation) and logarithmic velocity zone (above the vegetation). Both conditions combined, after some algebra and use of genetic programming, give the following expression for the Chézy coefficient induced by bottom and vegetation friction ($C$):

$$C = \left( \frac{1}{C_b^2} + \frac{C_D m D h}{2g} \right)^{-0.5} + \frac{\sqrt{g}}{\kappa} \ln \left( \frac{h}{H} \right)$$

where $C_b$ is the Chézy coefficient of the bed and $\kappa$ is the von Kármán constant. The corresponding Darcy-Weisbach friction

factor can be then obtained by

$$f = \frac{8g}{C^2}. \tag{2}$$

## 2.3  Järvelä

The approach from Järvelä (2004) was developed for flexible vegetation. It is based on the leaf area index (LAI), a dimensionless quantity that characterizes plant canopies. The LAI is defined as the one-sided leaf area per unit projected area in canopies.

The Darcy-Weisbach friction factor for vegetation ($f_v$) can be calculated by the following relation:

$$f_v = 4C_{D\chi}\text{LAI}\left( \frac{U}{U_\chi} \right)^\chi \frac{h}{H} \tag{3}$$

where $\chi$ is the species-specific vegetation parameter (Vogel exponent), $C_{D\chi}$ is the species-specific drag coefficient, $U$ is the mean flow velocity, $U_\chi$ is a normalizing value and is defined as the lowest flow velocity used in determining $\chi$. $U_\chi$ is usually 0.1 m/s and it will be considered constant.

## 20  2.4  Battiato and Rubol

The approach from Battiato and Rubol (2014) developed for submerged vegetation follows the concept of coupling an incompressible fluid flow with a porous medium flow. Although it is conceptually suited for rigid vegetation, this approach has been successfully validated also with flexible vegetation (see  Rubol et al., 2018). The main advantage of this approach lies in the representation of the drag force by a single parameter, i.e. the canopy permeability ($K$). The volumetric discharge per unit

width through a vegetated channel ($Q_w$) can be determined from direct integration of the velocity over depth, obtained from the solution of the coupled log-law and Darcy-Brinkman equations:

$$L = h - H, \quad \delta = LH^{-1}, \quad \lambda = HK^{-0.5}, \quad q = \rho g S_0 H^2 \mu_t^{-1}, \quad \mu_t = \rho \kappa' H u_*, \quad u_* = (g S_0 L)^{0.5}$$

$$C = 0.5\delta\lambda^{-1}\text{csch}(\lambda), \quad U^* = \lambda^{-2} + \delta\lambda^{-1}\text{coth}(\lambda)$$

$$Q_w = qH\{\lambda^{-2} + C\lambda^{-1}(e^\lambda - e^{-\lambda}) + \delta[(1+\delta)\ln(1+\delta) + U^* - \delta]\}$$





where $\rho$ is the density of water, $\mu_t$ is the turbulent viscosity, $\kappa'$ is the reduced von Kármán constant for vegetated channels ($\kappa' = 0.19$) and $u_*$ is the friction velocity. The Darcy-Weisbach friction factor can be then calculated by:

$$\tau_H = \rho g S_0 L, \quad U = Q_w h^{-1}$$

$$f = \frac{8\tau_H}{\rho U^2} \tag{4}$$

## 2.5 Overall comparison

From now on the presented floodplain friction formulations will be referred to as LIND, BAPT, JAER and BATT, respectively. The formulae will be analysed in terms of the total Darcy-Weisbach friction factor calculated as $f = f_b + f_v$, with $f_b$ and $f_v$ being the bottom and vegetation friction, respectively. The four expressions are then given by:

$$f = f_b + 4C_D m D h \tag{LIND}$$

$$f = 8g \left[ \left( \frac{1}{C_b^2} + \frac{C_D m D h}{2g} \right)^{-0.5} + \frac{\sqrt{g}}{\kappa} \ln\left( \frac{h}{H} \right) \right]^{-2} \tag{BAPT}$$

$$f = f_b + 4C_D \text{LAI} \left( \frac{U}{U_\chi} \right)^\chi \frac{h}{H} \tag{JAER}$$

$$f = 8g \frac{S_0 H}{U^2} \left( \frac{h}{H} - 1 \right) \tag{BATT}$$

In LIND and BAPT there is a direct dependency between the term $mD$ and the friction factor. The same analysis is valid for $C_D$ in the first three formulae. The relation $h/H$ is found in some form in all the approaches which include submerged vegetation. Furthermore, a similar relation between the bottom friction $C_b$ and the friction factor in BAPT is also observed in BATT. While the first three approaches present an explicit term for the bottom friction, in BATT the expression can be rearranged so that a Chézy-like term is found as a function of $H$.

## 3 Uncertainty analysis (UA)

Numerical models represent only an approximation of the observed process. The measured difference between the model and the observation can be considered either as error or uncertainty. Walters and Huyse (2002) defined these two concepts as:

- *Error*: a noticeable lack in the modelling process, not due to a lack of knowledge; (Deterministic)

- *Uncertainty*: a potential shortcoming in the modelling process due to a lack of knowledge. (Stochastic)

Uncertainty analysis aims to describe the system reliability by combining the uncertainties in the basic components (variables) of the system. The framework of the numerical model used to represent the system characterizes the interactions of the basic components. The overall response of the system is described by the performance function $Y$:

$$Y = f(x_1, x_2, \ldots, x_n) = f(\boldsymbol{x}) \tag{5}$$





where $x$ is the vector of input variables of the system and $n$ is the number of variables.

The analysis yields the combined effect of all input variables that significantly contribute to the performance function. The results from the analysis can be represented in terms of *reliability* or *risk*. Reliability refers to a prediction interval (PI), i.e. the probability that $Y$ will be found in the interval $[Y_a, Y_b]$. PI is expressed as the difference $|Y_a - Y_b|$ corresponding to a desired

probability. Risk refers to the probability of failure ($P_f$) with respect to a threshold value $Y_c$, i.e. the probability that $Y > Y_c$. $P_f$ is directly expressed as the calculated probability.

Three probabilistic methods have been chosen for the UA: First-Order Second-Moment, Monte Carlo (MC) and Metamodelling. The first method is based on the method of moments and requires the calculation of the model sensitivities (first-order derivatives). The MC method requires the simulation of a large number of random experiments and is the most expensive in

terms of computing time. The metamodelling method is based on random experiments (MC) with the benefit that it requires far less samples. Polynomial Chaos is a type of metamodelling technique, which is chosen for the present study. Further details on each method will be given in the following sections.

### 3.1 First-Order Second-Moment

Moment method approximations are obtained from the truncated Taylor series expansion about the expected value of the

input parameter. The First-Order Second-Moment (FOSM) method uses the first-order terms of the series and requires up to the second moments of the uncertain input variables for estimating the output variance of a system. The variance of the performance function $\sigma^2(Y)$ is given by:

$$\sigma^2(Y) = \sum_{i=1}^{n} \left[ \left( \frac{\partial f}{\partial x_i} \right)^2 \sigma^2(x_i) \right] \tag{6}$$

It should be noted that the FOSM method is suited as long as (a) the input variables are statistically independent and (b)

the linearity assumption is valid, i.e. the first-order approximation is enough to describe the sensitivity of the system. If $Y$ is non-linear, e.g. hydro- and morphodynamic models, one should make sure that the value of $\sigma(x_i)$ is small. Otherwise, $Y$ might be over- or underestimated. The reader is referred to Dettinger and Wilson (1981); Yen et al. (1986); Sitar et al. (1987) for further details on FOSM.

### 3.2 Monte Carlo

Monte Carlo simulation is a probabilistic method in which a very large number of similar random experiments form the basis. An attempt is made to solve analytically unsolvable or complicated solvable problems with the help of probability theory. The law of large numbers makes up one of the main aspects of the method. The random experiments can be carried out in computer calculations in which (pseudo)random numbers are generated with suitable algorithms to simulate random events.

The basic steps of a MC method can be described as follows:

1. Sample the input random variables $x$ from their known or assumed probability density function $N$ times;

    2. Calculate the deterministic output $Y$ for each input sample;



3. Determine the statistics of the distribution of $Y$ (e.g. mean, variance).

Step (2) should be repeated $N$ times, which presents this method's main drawback. Also the input variables are considered to be statistically independent, otherwise the joint probability distribution is required. The advantage is its robustness, because independently from the nature of $Y$ (linear or non-linear), the method will always deliver reliable results as long as the number of samples ($N$) is sufficiently large.

## 3.3 Metamodelling

Metamodelling attempts to offset the increased cost of probabilistic modelling by replacing the expensive evaluation of model calculations with a cost-effective evaluation of surrogates. Polynomial Chaos (PC) is a powerful metamodelling technique that aims to provide a functional approximation of a computational model through its spectral representation of uncertainty based on polynomial functions. A more detailed introduction to the PC method can be found in Marelli and Sudret (2017).

Spectral-based methods allow for an efficient stochastic reduced basis representation of uncertain parameters in numerical modelling. By means of a truncated expansion to discretize the input random quantities it is possible to reduce the order of complexity of the system. Let us consider the uncertain parameter $A$, representing velocity, density, or pressure in a stochastic fluid dynamics problem, as:

$$A(\boldsymbol{x},t,\boldsymbol{\xi}) \approx \sum_{j=0}^{P} a_j(\boldsymbol{x},t)\Psi_j(\boldsymbol{\xi}) \tag{7}$$

where $a_j(x,t)$ is the deterministic component, $\Psi_j(\boldsymbol{\xi})$ is the random basis function corresponding to the $j$-th mode and $\boldsymbol{\xi}$ is the random variable vector characterizing the uncertainty in the parameter. The polynomial chaos expansion in (7) is approximated by a discrete sum taken over the number of output modes $P$ defined as:

$$P+1 = \frac{(n+d)!}{n!d!} \tag{8}$$

where $d$ is the degree of the polynomial and $n$ is the number of random dimensions. The statistics of the distribution for the model output at a specific position and time can be calculated using the coefficients and the basis functions. The mean and variance of the solution is given respectively by

$$E[A(\boldsymbol{x},t,\boldsymbol{\xi})] = \int_R A(\boldsymbol{x},t,\boldsymbol{\xi})p(\boldsymbol{\xi})d\boldsymbol{\xi} = a_0(\boldsymbol{x},t) \tag{9}$$

$$\mathrm{Var}[A(\boldsymbol{x},t,\boldsymbol{\xi})] = \int_R [A(\boldsymbol{x},t,\boldsymbol{\xi}) - a_0(\boldsymbol{x},t)]^2 p(\boldsymbol{\xi})d\boldsymbol{\xi} \tag{10}$$

with $p(\boldsymbol{\xi})$ being the weight function of the polynomial and $R$ its the support range. When the input uncertainty is Gaussian (normal) the basis function $\Psi(\boldsymbol{\xi})$ takes the form of a multi-dimensional Hermite Polynomial, so that $R = (-\infty, +\infty)$.

In this study, the Non-Intrusive Polynomial Chaos method (NIPC) will be considered. The main objective of this method is to obtain the polynomial coefficients without modifying the original model. This approach considers the deterministic model





as a "black-box" and approximates the polynomial coefficients based on model evaluations. The advantage is that this method requires much fewer evaluations of the original model (with regard to MC) for providing reliable results (at least one order of magnitude). The main disadvantage is that it is an additional approximation in the modelling framework, thus leading to further loss of information of the physical process. The reader is referred to Hosder and Walters (2010) for further details on

the application of the NIPC method. The implementation of the method was done in *Python* with the help of the OpenTURNS package (Baudin et al., 2015).

## 4   Case study

The current study focuses on a reach of the river Rhine used for numerical tests by the German Federal Waterways Engineering and Research Institute (BAW). It is an 11 km long section of the lower Rhine located between kilometres 738 and 750, nearby

Düsseldorf (Germany). The model has been extensively tested and calibrated for a wide spectrum of discharges. A constant discharge of 7870 m$^3$/s was imposed at the upstream boundary and the corresponding free surface at the downstream boundary. These conditions represent a flood scenario with a probability of occurrence larger than HQ$_5$ (LUA, 2002). In recent years flood studies are receiving more and more attention as part of BAW's activities. For that reason the current motivation is to understand how sensitive numerical models are to floodplain friction under flood conditions and how this might affect the hydrodynamics

of navigation channels. An overview of the study area is presented in Figure 1, where the red polygon delimits the boundaries of the numerical model.

### 4.1   Hydrodynamic model

A numerical model is used to simulate the flood scenario. In the BAW studies carried out in large scale river projects ($10^1$-$10^2$ km) usually make use of the hydrodynamic model TELEMAC-2D (Galland et al., 1991; Hervouet and Ata, 2017). It is a

two-dimensional finite-element software (finite-volume also available) for solving the shallow water equations, a set of partial differential equations derived from the integration of the Navier-Stokes equations over the vertical axis. Thus, the equations for the conservation of mass and momentum in two dimensions should be solved.

$$\frac{\partial h}{\partial t} + \frac{\partial uh}{\partial x} + \frac{\partial vh}{\partial y} = 0 \tag{11}$$

$$\frac{\partial u}{\partial t} + u\frac{\partial u}{\partial x} + v\frac{\partial u}{\partial y} = -g\frac{\partial(z_b + h)}{\partial x} + \nu\left(\frac{\partial^2 u}{\partial x^2} + \frac{\partial^2 u}{\partial y^2}\right) - \frac{\tau_x}{\rho h} + S_x \tag{12}$$

$$\frac{\partial v}{\partial t} + u\frac{\partial v}{\partial x} + v\frac{\partial v}{\partial y} = -g\frac{\partial(z_b + h)}{\partial y} + \nu\left(\frac{\partial^2 v}{\partial x^2} + \frac{\partial^2 v}{\partial y^2}\right) - \frac{\tau_y}{\rho h} + S_y \tag{13}$$

where $h$ is the water depth, $z_b$ is the bottom elevation, $u, v$ are the components of the velocity field, $\nu$ is the fluid viscosity, which may be constant or given by a turbulence model, $\tau_x, \tau_y$ are the shear stress components and $S_x, S_y$ are any additional source term components of momentum (e.g. wind stress, external forces). The bottom shear stress is bound to the depth-averaged



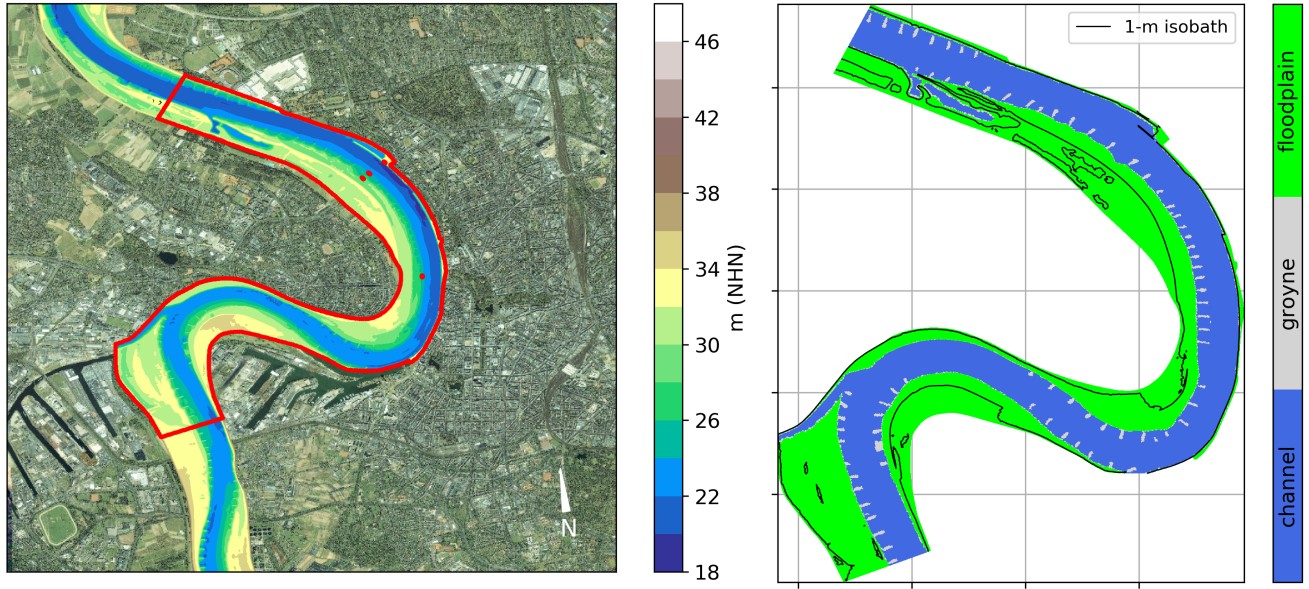

**Figure 1.** Lower Rhine river topography and numerical model boundaries (red polygon) nearby Düsseldorf (left), and friction zones definition in the numerical model (right). The flow direction is north. [Satellite image (left) is the intellectual property of Esri and is used herein under license. Copyright © 2018 Esri and its licensors.]

velocity by the quadratic law first introduced by Taylor and Shaw (1920):

$$\tau = c_f \rho \boldsymbol{u} |\boldsymbol{u}| \tag{14}$$

$$c_f = c_{fb} + c_{fv} \tag{15}$$

The friction coefficient ($c_f$) is equal to the sum of the bottom friction ($c_{fb}$) and the friction due to vegetation ($c_{fv}$). The

5  bottom friction usually can be determined by traditional friction laws relating open-channel flow velocity to resistance coefficient (e.g. Manning, Chézy, Darcy-Weisbach, Nikuradse). However, on floodplains the velocity profile strongly depends on the vegetation height and morphology. Thus, specific flow resistance formulae have been developed for determining the vegetation drag (see Section 2).

The model consists of an unstructured triangular mesh composed by 56825 points and 112360 elements. The resolution

10  varies from about 2.5 m in the main channel to about 30 m on the floodplains and the model mesh covers an area of ca. 8 km$^2$. A constant discharge upstream and a constant water level downstream are imposed at the open boundaries, as aforementioned. A time step of 1 s guarantees a Courant number below 1 and it is used to simulate 24 hours, which takes about 9 min with the LIND formulation using 160 processors of the BAW's HPC. The other three formulations for floodplain friction are about three times faster (3.5 min) as there is no iteration step.

15  This numerical model has been extensively investigated from the point of view of sediment transport and morphodynamics (Backhaus et al., 2014; Riesterer et al., 2014). Because water level measurements along the river channel axis are available





for a discharge of 7870 m$^3$/s, the bottom friction in the numerical model has been calibrated under these conditions as a representation of a flood scenario. The bottom friction in the model defined by Nikuradse's equivalent sand roughness ($k_s$) in the channel is set to 0.1 m. Originally the floodplains are divided basically in three categories: forest, cultivated land and meadows/pastures. In the current study, however, all the floodplain areas are considered to be covered with the same type of vegetation for the sake of simplicity (Figure 1).

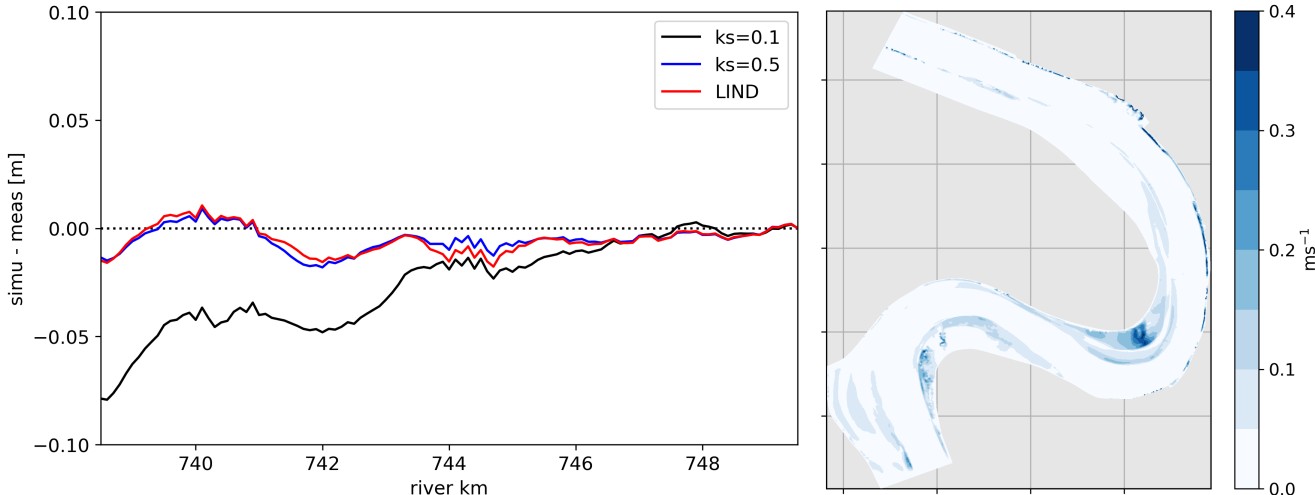

**Figure 2.** Water level difference along the river channel axis for different floodplain friction values (left), and absolute difference of the flow velocity with and without floodplain friction formulation (right).

It is possible to calibrate the floodplain friction in the hydrodynamic model to fit water level measurements either with the traditionally used Lindner-Pasche friction law (1) in addition to the bottom friction (Figure 2l, red line), or only a higher bottom friction value on the floodplains (Figure 2l, blue line). If the same friction value of the river channel ($k_s = 0.1$ m) is used for the floodplains, the momentum is too high and the simulated free surface does not fit the measurements (black line). A much

10   better result is obtained by using $k_s = 0.5$ m on the floodplains (blue line), in which the RMSE is reduced to less than 25% from the first result. Alternatively, a similar result is obtained when 5 cm thick stems evenly spaced by 5 m intervals are added to the floodplains with the LIND formulation, while keeping the bottom friction equal to the channel.

The reader may ask himself/herself about which approach to be used. In this case it is useful to compare the absolute difference of the flow velocity with and without the floodplain friction formulation (see Figure 2r). It can be seen that while

15   differences in the main channel can be neglected ($< 0.05$ m/s), those on the floodplains cannot (up to 0.4 m/s). In other words, when model calibration is based only on measurements in the river channel the hydrodynamics on floodplains is not guaranteed to be correctly simulated. It is important to point out that in case of unsteady flow conditions a friction formulation dependent on water depth is not desired. However, if flow velocity measurements are not available on the floodplains (usually the case), little can be done in terms of calibration. Furthermore, remote sensing data of vegetation characteristics (Light Detection And



Ranging technology) have been used in flood modelling in the last 20 years, but the accuracy of these measurements should also be taken into account (see Cobby et al., 2003; Antonarakis et al., 2008; Dombroski, 2017).

An alternative to the deterministic approach in such situations, when there is a potential shortcoming in the modelling process due to a lack of information, is to carry out an UA. As explained in Section 3, this analysis can be used to determine the combined effect of all uncertain input parameters that significantly affect model results by means of a probabilistic approach.

## 4.2 Input parameters

The next step now is to calibrate the remaining floodplain friction formulations with water level measurements. In order to make a comparison to the LIND approach, first $H = 10.0$ m is set to ensure emergent conditions in all formulations. A second scenario is then calibrated for submerged conditions, in which $H = 1.0$ m. Tables 1 and 2 present the calibrated parameters for each one of the scenarios. After the calibration all model results presented RMSE smaller than 1 cm for water level. (The density of stems $m$ is calculated as $1/d^2$, where $d$ is the distance between stems.)

**Table 1.** Floodplain friction parameter values calibrated under emergent conditions ($H = 10.0$ m).

| floodplain friction formulation | $D$ [m] | $d$ [m] | $C_D$ [-] | LAI [-] | $\chi$ [-] | $K$ [m$^2$] |
|---|---|---|---|---|---|---|
| LIND | 0.05 | 5.0 | - | - | - | - |
| BAPT | 0.05 | 2.5 | 0.5 | - | - | - |
| JAER | - | - | 0.5 | 0.5 | -0.9 | - |
| BATT | - | - | - | - | - | 0.02 |

**Table 2.** Floodplain friction parameter values calibrated under submerged conditions ($H = 1.0$ m).

| floodplain friction formulation | $D$ [m] | $d$ [m] | $C_D$ [-] | LAI [-] | $\chi$ [-] | $K$ [m$^2$] |
|---|---|---|---|---|---|---|
| BAPT | 0.01 | 0.333 | 0.25 | - | - | - |
| JAER | - | - | 0.25 | 0.3 | -0.9 | - |
| BATT | - | - | - | - | - | 0.9 |

For the UA it is required that all sensitive parameters relevant to model results should be considered for the determination of the prediction intervals (PI). Once the parameters are chosen a very important step follows: an error or deviation should be carefully assigned to each parameter. This variation should be small enough to be treated as an error, but large enough to include the actual parameter uncertainty (due to the lack of knowledge). Unfortunately there is no general rule for choosing a proper value, since different aspects might contribute e.g. measurement accuracy, spatial/time variances, numerical representation of process, etc. In the current study, the chosen variations for the parameters related to the vegetation species ($C_D$, LAI, $\chi$)





are based on values given in Aberle and Järvelä (2013); Västilä and Järvelä (2014). For the remaining parameters ($H$, $D$, $d$, $K$) a standard deviation of 10% of the calibrated value is assumed ($\sigma = 0.1\mu$). The vegetation height $H$ is only included in the analysis under submerged conditions. Input deviations are treated as errors and, therefore, represented by a Gaussian distribution. This implies that there is a 99.7% probability that the parameter value is found within $[\mu - 3\sigma, \mu + 3\sigma]$.

**Table 3.** Input parameters deviations and ranges.

| parameter | $H^\star$ [m] | $D$ [m] | $d$ [m] | $C_D$ [-] | LAI [-] | $\chi$ [-] | $K$ [m$^2$] |
|---|---|---|---|---|---|---|---|
| $\sigma$ | 0.1 | $0.1\mu$ | $0.1\mu$ | 0.05 | 0.033 | 0.033 | $0.1\mu$ |
| min | 0.01 | 0.001 | 0.1 | 0.1 | 0.2 | -1.0 | 0.001 |
| max | 10.0 | 1.0 | 100.0 | 0.7 | 3.2 | -0.4 | 10.0 |

$^\star$ only for submerged conditions

Finally, the UA methods presented in Section 3 are applied with the input uncertainties given in Table 3. The FOSM method is evaluated through central finite difference; hence, $2n$ model evaluations are necessary ($n$ refers to the number of input variables). For the MC method, a sample size of 1000 has been used for the evaluation. Although MC sample sizes are usually considered in the range of $10^4$-$10^5$, previous tests with $10^4$ samples showed very little difference in results. The NIPC method (metamodelling) requires less samples than MC, because a polynomial function fitted to the samples is then used for the evaluation of results. In this case results with 100 samples for the metamodel have been sufficient for approximating MC results.

## 5   Results and Discussion

The numerical model has been evaluated with the four floodplain friction formulations. A constant discharge of $7870\,\mathrm{m^3/s}$ was imposed at the upstream boundary and at the downstream boundary a corresponding free surface based on a discharge curve. Model results have been analysed after a steady state was achieved in the simulation and presented in the form of prediction interval (PI) with a 95%-probability of occurrence. It should be noted that the PI represents a range of variation around the mean value, which is not necessarily symmetric (MC and metamodelling).

In Figure 3 the uncertainty analysis of the flow velocity under emergent vegetation conditions is presented. It can be observed that similar results are obtained with the three UA methods. Among the friction formulations, LIND and JAER presented smaller variations on average and the PI exceeds $0.2\,\mathrm{m/s}$ only at the left floodplain in the middle of the river reach. On the other hand the BATT approach appears to be the most sensitive one, followed by BAPT. The approach from Battiato and Rubol results in PI $> 0.2\,\mathrm{m/s}$ on most of the floodplains. Relative to results with calibrated values all formulations showed variations above 10% on the floodplains. The same analysis is carried out for submerged conditions (see Figure 4). Because the LIND formulation is only valid for emergent conditions (independent of $H$), submerged conditions cannot be accounted for.



As expected all results present on average larger PI than under emergent conditions, due to the addition of $H$ in the analysis. The floodplain PI of flow velocity is mostly above $0.2\ \mathrm{m/s}$ and in BATT the $\mathrm{PI} > 0.5\ \mathrm{m/s}$ is present on floodplains located at the inner bends of the river reach. In the channel the PI in BATT is the largest one and exceeds $0.05\ \mathrm{m/s}$ all along the upstream river section. Relative to results with calibrated values the variations exceed now 25% on the floodplains, and in BATT at

shallow regions up to 100%.

As explained in Section 3 results from the uncertainty analysis can also be represented in terms of risk, i.e. a probability of failure ($P_f$). This is a more suitable analysis for when results must not exceed a given threshold. For instance, a threshold of $0.1\ \mathrm{m/s}$ above the mean value is used for the analysis of the flow velocity (see Figure 5). In other words, the probability of exceedance of $\bar{u} + 0.1\ \mathrm{m/s}$ was calculated. Because in the current study the difference among the UA methods was not

significant, results are now presented only from metamodelling. Results indicate that there is a larger probability that velocities are found above $\bar{u} + 0.1$ with the BATT approach. Under submerged conditions velocities are more likely to exceed this threshold. In BAPT and BATT the probability of failure can be higher than 10% on the floodplains.

Figures 3 and 4 show that results using the UA methods are similar for this case study. Although the FOSM method is the less expensive alternative among the ones presented (only $2n$ model evaluations), it should be used with caution as it is a first-

order method (linear approximation). In studies involving strongly dominant non-linear processes (e.g. turbulence modelling, sediment transport, unsteady conditions, etc.) further comparisons should be carried out. On the other hand, Monte-Carlo based methods have the advantage that the analysis under any conditions is possible. Although a large number of simulations is required for obtaining trustful results, alternatives such as the NIPC make them more feasible by reducing the sample size by at least one order of magnitude. For instance, Mouradi et al. (2016) carried out the UA of a computationally intensive

morphodynamic model, to which they applied pure MC and metamodelling methods.

When compared to emergent conditions the overall uncertainty of submerged conditions is significantly larger. This is an expected result in uncertainty analysis as there is an additional input (vegetation height $H$) that significantly contributes to model performance. The floodplain friction formulation Lindner-Pasche is by definition only valid for emergent conditions. Thus, a different approach is needed when submerged conditions should be taken into account. Additionally, Brandimarte and

Di Baldassare (2012) warn that when simulating flood scenarios attention must be given to parameter compensation between floodplain and channel resistance coefficients, so that reasonable values are chosen.

An important topic not only regarding uncertainty analysis but numerical simulation in general, is the matter of input uncertainty definition. When performing a numerical simulation that is based on physical processes one will eventually need to validate calculations with measurements. Also, initial and boundary conditions usually are based on measurements of the orig-

inal process. That is to say one should know a priori how accurate the available measurements are. This is usually not a trivial task, since measurement errors may not be easily evaluated (see Taylor, 1997). For instance, Di Baldassarre and Montanari (2009) published a study that focused only on the uncertainty in river discharge observations. Although it was attributed a standard deviation of 2.7% for discharge measurement errors, the authors emphasize that this value is associated to their case study, thus any generalization should be attributed with care. For uncertainties related to floodplain friction there are no such

reference studies known to the authors. In that case, a suggested practice is to start with commonly used value ranges in the



**Figure 3.** Prediction interval of flow velocity under emergent conditions (rows: friction formulations; columns: UA methods).




**Figure 4.** Prediction interval of flow velocity under submerged conditions (rows: friction formulations; columns: UA methods).





**Figure 5.** Risk analysis of flow velocity for threshold $\bar{u} + 0.1$ m/s under emergent (top) and submerged (bottom) conditions.

literature and apply a six sigma range ($[\mu - 3\sigma, \mu + 3\sigma]$) for the total parameter variation as a rule of thumb. Of course available experience in the topic of investigation should be also taken into account.

# 6   Conclusions

A framework for the estimation of uncertainties of hydrodynamic models on floodplains was presented. A traditional resistance
formula used for river modelling together with three more recent approaches to floodplain friction were considered for carrying
out an uncertainty analysis. The analysis was performed by means of three different methods: traditional MC, FOSM and NIPC





(metamodelling). A two-dimensional model of a 10 km reach of the River Rhine was calibrated under steady flow conditions and the analysis was based on flow velocity results.

The tested floodplain friction formulae produced qualitatively similar results, in which uncertainties in flow velocity are most significant where the resistance coefficient was modified. Under emergent conditions, larger velocity variations are obtained with the formulations of BAPT and BATT. Variations from the latter also included the river channel. Under submerged conditions all approaches resulted in larger uncertainties, as the vegetation height has been included in the analysis. Although the BATT approach presented once again the largest variations among the analysed methods, results were consistent not only qualitatively, but also quantitatively. In summary, among the tested floodplain friction formulae the JAER approach presented on average the smallest prediction intervals i.e. the most accurate results. It is important to keep in mind that UA results are not only dependent on the defined input parameters deviations, but also on the number of parameters considered in the analysis. In that sense, the BATT approach is still attractive for it reduces the current analysis to a single parameter, the canopy permeability $K$.

The three UA methods compared gave similar results, which means that FOSM is the most efficient in this case. Despite being a very simple method to apply, FOSM will only produce good results when the first-order approximation is sufficient to describe the sensitivity of the system. In the presented study this was the case, probably because all the chosen inputs are directly correlated to the resistance coefficient. Research on related topics such as floodplain mapping usually focuses on the analysis of uncertainties that relies on Monte-Carlo based methods (e.g., Di Baldassarre et al., 2010; Domeneghetti et al., 2013). Several further topics could be listed here for future development e.g. unsteady flow, boundary conditions, not to mention sediment transport modelling. However, the most important is first to be aware of the limitations of the available information and tools. Are there enough measurements for an acceptable calibration in the study area? Is the chosen numerical model capable of correctly representing the physical process under the desired conditions? As basic as it may sound, if those questions cannot be answered, any kind of analysis involving uncertainties will fail in providing useful results.

*Competing interests.* The authors declare that no competing interests are present.

*Acknowledgements.* The authors would like to thank the EDF researcher Cédric Goeury for his help with the metamodel setup, and Audrey Valentine for her valuable comments in revising this manuscript.





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
