# Peer review of "Uncertainty quantification of floodplain friction in hydrodynamic models"

_Hydrology and Earth System Sciences, 2019_

## Referee Comment (RC1) · Renata Romanowicz (Referee) · 3 May 2019

The authors present an interesting study of the variability of the velocity predictions of a 2-D hydrodynamic model related to the uncertainty of vegetated floodplain friction parameterisation. Four parameterisation models are tested using three uncertainty analysis methods. These methods include First Order Second Moment, Monte Carlo sampling and metamodeling (Non-Intrusive Polynomial Chaos).

My main concern is the misleading formulation of the problem. Namely, the authors introduce the term 'uncertainty' based on the error between simulated and observed variables (page 5, lines 19-26) and apply it to analysing the variability of the model output in the form of velocity simulations. In other words, a sensitivity analysis is performed

instead of the earlier-defined 'uncertainty' analysis. Unfortunately, the wrong use of the term 'uncertainty' leads to wrong conclusions. Four different friction parameterisation models have different numbers of parameters (from one to three). According to the authors, the model with three parameters shows smaller uncertainty than the model with only one parameter. This could be true only when the term 'uncertainty' is replaced by 'variability'. It simply shows that some parameters in that particular parameterisation scheme have a small influence on model output. Unfortunately, it does not mean that the model output has a small uncertainty (i.e. is better defined). In some way, the authors do the opposite to what Gupta and Razavi (2018) described as a sensitivity analysis using the goodness of fit criterion instead of output variables. The latter and the present papers show that a clear formulation of the problem helps to avoid drawing wrong conclusions.

In summary, the authors are asked to correct their problem formulation and apply a sensitivity method (e.g. the Global Sensitivity Analysis GSA of Saltelli et al., 2004). Berends et al. (2018) could also be helpful in dealing with the high computer time costs of hydraulic models. My specific comment regards the calibration method which is not explained.

References

Berends, K.D., J.J. Warmink, S.J. M.H. Hulscher, 2018, Efficient uncertainty quantification for impact analysis of human interventions in rivers, Environmental Modelling & Software, 107, 50-58 https://doi.org/10.1016/j.envsoft.2018.05.021.

Gupta, H.V. and S. Razavi, Revisiting the basis of sensitivity analysis for dynamical Earth System models, 2018, Water Resources Research, https://doi.org /10.1029/2018WR022668.

Saltelli, A., S. Tarantola, F. Campolongo, M. Ratto, 2004, Sensitivity analysis in practice: a guide to assessing scientific models, Chichester, Wiley.

---

## Author Comment (AC1) · 15 May 2019

Dear Prof. Renata Romanowicz,

thank you very much for your valuable comments to our article. We read them carefully and addressed them in the following text. An updated version of the manuscript including your suggestions is currently being prepared and will be soon available.

The use of the terms "uncertainty" vs. "sensitivity" analysis seems to be a constant discussion in the scientific community and it obviously leads to misunderstandings. For example in the references you mentioned: Saltelli et al. (2004) wrote (Box 1.1) "This is in fact an uncertainty analysis, e.g. a characterisation of the output distribution of Y given the uncertainties in its input."; Berends et al. (2018) used the Monte Carlo

method and referred to the results as "uncertainty estimation /quantification"; Saltelli et al. (2008) in Section 1.1.4 described exactly what we presented in our analysis with the Monte Carlo method as "uncertainty analysis". Further examples of the use of the term "uncertainty analysis" can be seen in Hofer (1999), Maskey and Guinot (2003) and Altarejos- García et al. (2012), where the term was employed similarly to the way we did. Furthermore, Walters and Huyse (2002) described in Section 2 ("Review of Uncertainty Analysis Methods") amongst others the same three methods we used.

I understand the need for a common language and agreement in using identical names when addressing identical things. Therefore, my suggestion would be to exchange the term "uncertainty analysis" with "uncertainty quantification" in our manuscript. This would be in agreement with Berends et al. (2018) and with other studies carried out similarly to ours, e.g. Hosder and Walters (2010), Oladyshkin and Nowak (2012), and Sudret (2015).

Our goal of investigation is to quantify the uncertainties of hydrodynamic model results on floodplains with regard to different friction methods. Within the large number of different friction methods there is still no generally accepted method for large scale applications. The outcomes of the uncertainty quantification will help to choose a better suited friction method for practical use. The model was previously calibrated based on the best information available and the input parameters are perturbed within a practical range of variation, and not across the whole feasible parameter space. Analyses considering the entire parameter space are still computationally unfeasible in real engineering projects involving large models and cannot be put in practice in our case.

With respect to the problem formulation we will improve the description in Sections 1 and 3 accordingly. Furthermore, from the sensitivity methods presented in Saltelli et al. (2004), we will add scatterplots and calculate the standardised regression coefficient (SRC) to assist the evaluation of each friction formulation (see figures). With respect to the calibration method, we will emphasize in Section 4.1 the fact that previous investigations already presented good results for the hydrodynamics. This knowledge was

the starting point for our study.

References

Altarejos-García, L., Martínez-Chenoll, M. L., Escuder-Bueno, I., and Serrano-Lombillo, A.: Assessing the impact of uncertainty on flood risk estimates with reliability analysis using 1-D and 2-D hydraulic models, Hydrol. Earth Syst. Sci., 16, 1895-1914, 2012.

Berends, K. D., Warmink, J. J., Hulscher, S. J. M. H. Efficient uncertainty quantification for impact analysis of human interventions in rivers, Environmental Modelling & Software, 107, 50-58, 2018.

Hofer, E. Sensitivity analysis in the context of uncertainty analysis for computationally intensive models, Computer Physics Communications, 117, 21-34, 1999.

Hosder, S. and Walters, R.: Non-Intrusive Polynomial Chaos Methods for Uncertainty Quantification in Fluid Dynamics, in: Proc. 48th AIAA Aerospace Sciences Meeting Including the New Horizons Forum and Aerospace Exposition, 2010. Maskey, S. and Guinot, V. Improved first-order second moment method for uncertainty estimation in flood forecasting, Hydrological Sciences Journal, 48(2), 183-196, 2003.

S. Oladyshkin, W. Nowak, Data-driven uncertainty quantification using the arbitrary polynomial chaos expansion, Reliability Engineering & System Safety, 106, 179-190, 2012.

Saltelli A., Ratto, M., Andres, T., Campolongo, F., Cariboni, J., Gatelli, D., Saisana, M. and Tarantola, S. Global sensitivity analysis: the primer, John Wiley, 2008.

Saltelli, A., Tarantola, S., Campolongo, F., Ratto, M. Sensitivity analysis in practice: a guide to assessing scientific models, Wiley, 2004.

Sudret, B. Polynomial chaos expansions and stochastic finite element methods, In: Risk and Reliability in Geotechnical Engineering (Chap. 6), pp. 265-300, CRC Press,

2014.

Walters, R. and Huyse, L. Uncertainty analysis for fluid mechanics with applications, Tech. Rep. 2002-1, Institute for Computer Applications in Science and Engineering, Hampton: ICASE, NASA Langley Research Center, 2002.

[Figure]

[Figure]

**Fig. 1.** Scatterplot of scalar velocity for BAPT friction formulation.

[Figure]

**Fig. 2.** Scatterplot of scalar velocity for JAER friction formulation.

[Figure]

**Fig. 3.** Scatterplot of scalar velocity for BATT friction formulation.

---

## Referee Comment (RC2) · Renata Romanowicz (Referee) · 2 Jun 2019

The authors have improved on the clarity of the description of the methods. The sensitivity plots are a good illustration of the approach taken. A minor comment: the velocity units (y-axis) in Fig. 3 are missing. Also, the linear relationship between flow velocity and a canopy permeability K requires a comment.

I understand that different definitions of the same word (sensitivity vs uncertainty) are used in different disciplines. As long as those definitions are clearly stated and we know what the discussion is about, it does not make much difference to me.

However, the statement "... the smallest prediction intervals, i.e. the most accurate results" (line 9, page 1 and line 24, page 18) in the absence of observations is not

justified. It should be replaced by: . . . "the smallest variance".

---

## Referee Comment (RC3) · Anonymous Referee #2 · 5 Jun 2019

Overview This study describes an interesting analysis on the estimation of uncertainty of hydrodynamic models on floodplains. Specifically, the variability of the velocity predictions of a 2-D hydrodynamic model related to the uncertainty of vegetated floodplain friction parameterization is investigated. Four traditional floodplain resistance formulae are considered using three different uncertainty analysis (UA) methods: i.e. First Order Second Moment, Monte Carlo sampling and metamodeling. The analysis carried on a case study selected along the Rhine River, show that the three UA methods compared gave similar results which means that First-Order Second-Moment is the less expensive one.

Comments The topic of the work is of interest for the scientific community and consistent with the aim of the journal. English is sound and the manuscript is well written. I

was able to follow the analysis carried out by the authors even if I suggest some necessary modifications. One limit concerns the confusion on the used symbols: some of them are not clearly defined, both in the text and in the tables captions (e.g. H, D, t, x, y, ...), and some other are used to indicate more than one quantity (e.g., d). Moreover, the authors use acronyms before they are defined. I was wondering about the meaning of the term 'prediction interval' and if it is considered as an 'uncertainty band'. The comment of the previous reviewer and the reply of the authors shed light on this issue. I must say that, from my point of view, the term 'sensitivity analysis' would be more appropriate in this case. Minor comments: - explain what 'with a probability of occurrence larger than HQ5' means. - use always the past tense or the present tense throughout the manuscript.
* * *

---

## Author Response (AR1)

**RC1**

The authors present an interesting study of the variability of the velocity predictions of a 2-D hydrodynamic model related to the uncertainty of vegetated floodplain friction parameterisation. Four parameterisation models are tested using three uncertainty analysis methods. These methods include First Order Second Moment, Monte Carlo sampling and metamodeling (Non-Intrusive Polynomial Chaos).

My main concern is the misleading formulation of the problem. Namely, the authors introduce the term 'uncertainty' based on the error between simulated and observed variables (page 5, lines 19-26) and apply it to analysing the variability of the model output in the form of velocity simulations. In other words, a sensitivity analysis is performed instead of the earlier-defined 'uncertainty' analysis. Unfortunately, the wrong use of the term 'uncertainty' leads to wrong conclusions. Four different friction parameterisation models have different numbers of parameters (from one to three). According to the authors, the model with three parameters shows smaller uncertainty than the model with only one parameter. This could be true only when the term 'uncertainty' is replaced by 'variability'. It simply shows that some parameters in that particular parameterisation scheme have a small influence on model output. Unfortunately, it does not mean that the model output has a small uncertainty (i.e. is better defined). In some way, the authors do the opposite to what Gupta and Razavi (2018) described as a sensitivity analysis using the goodness of fit criterion instead of output variables. The latter and the present papers show that a clear formulation of the problem helps to avoid drawing wrong conclusions.

In summary, the authors are asked to correct their problem formulation and apply a sensitivity method (e.g. the Global Sensitivity Analysis GSA of Saltelli et al., 2004). Berends et al. (2018) could also be helpful in dealing with the high computer time costs of hydraulic models. My specific comment regards the calibration method which is not explained.

**Dear Prof. Renata Romanowicz,**

thank you very much for your valuable comments to our article. We read them carefully and addressed them in the following text. An updated version of the manuscript including your suggestions is currently being prepared and will be soon available.

The use of the terms "uncertainty" vs. "sensitivity" analysis seems to be a constant discussion in the scientific community and it obviously leads to misunderstandings. For example in the references you mentioned: Saltelli et al. (2004) wrote (Box 1.1) "This is in fact an uncertainty analysis, e.g. a characterisation of the output distribution of Y given the uncertainties in its input."; Berends et al. (2018) used the Monte Carlo method and referred to the results as "uncertainty estimation /quantification"; Saltelli et al. (2008) in Section 1.1.4 described exactly what we presented in our analysis with the Monte Carlo method as "uncertainty analysis". Further examples of the use of the term "uncertainty analysis" can be seen in Hofer (1999), Maskey and Guinot (2003) and Altarejos- García et al. (2012), where the term was employed similarly to the way we did. Furthermore, Walters and Huyse (2002) described in Section 2 ("Review of Uncertainty Analysis Methods") amongst others the same three methods we used.

I understand the need for a common language and agreement in using identical names when

addressing identical things. Therefore, my suggestion would be to exchange the term "uncertainty analysis" with "uncertainty quantification" in our manuscript. This would be in agreement with Berends et al. (2018) and with other studies carried out similarly to ours, e.g. Hosder and Walters (2010), Oladyshkin and Nowak (2012), and Sudret (2015).

Our goal of investigation is to quantify the uncertainties of hydrodynamic model results on floodplains with regard to different friction methods. Within the large number of different friction methods there is still no generally accepted method for large scale applications. The outcomes of the uncertainty quantification will help to choose a better suited friction method for practical use. The model was previously calibrated based on the best information available and the input parameters are perturbed within a practical range of variation, and not across the whole feasible parameter space. Analyses considering the entire parameter space are still computationally unfeasible in real engineering projects involving large models and cannot be put in practice in our case.

With respect to the problem formulation we will improve the description in Sections 1 and 3 accordingly. Furthermore, from the sensitivity methods presented in Saltelli et al. (2004), we will add scatterplots and calculate the standardised regression coefficient (SRC) to assist the evaluation of each friction formulation (see figures). With respect to the calibration method, we will emphasize in Section 4.1 the fact that previous investigations already presented good results for the hydrodynamics. This knowledge was the starting point for our study.

**RC2**

The authors have improved on the clarity of the description of the methods. The sensitivity plots are a good illustration of the approach taken. A minor comment: the velocity units (y-axis) in Fig. 3 are missing. Also, the linear relationship between flow velocity and a canopy permeability K requires a comment.

I understand that different definitions of the same word (sensitivity vs uncertainty) are used in different disciplines. As long as those definitions are clearly stated and we know what the discussion is about, it does not make much difference to me.

However, the statement "... the smallest prediction intervals, i.e. the most accurate results" (line 9, page 1 and line 24, page 18) in the absence of observations is not justified. It should be replaced by: ... "the smallest variance".

**Dear Prof. Renata Romanowicz,**

thank you very much for your comments. I think the scatterplots improved the understanding of model behaviour in our analysis and could be a good starting point for further investigations. The missing units in Figure 3 were added. Regarding the analysis of the canopy permeability, a missing description of the approach was added in Section 2.4 and referenced in the discussion. I hope it is clear now. You are correct about our statement on the prediction intervals. It was suppose to be "most precise" and not "most accurate", of course. Nevertheless I corrected it as you suggested. Please find the corrected version of the manuscript attached as supplement.

**RC3**

Overview This study describes an interesting analysis on the estimation of uncertainty of hydrodynamic models on floodplains. Specifically, the variability of the velocity predictions of a 2-D hydrodynamic model related to the uncertainty of vegetated floodplain friction parameterization is investigated. Four traditional floodplain resistance formulae are considered using three different uncertainty analysis (UA) methods: i.e. First Order Second Moment, Monte Carlo sampling and metamodeling. The analysis carried on a case study selected along the Rhine River, show that the three UA methods compared gave similar results which means that First-Order Second-Moment is the less expensive one.

Comments The topic of the work is of interest for the scientific community and consistent with the aim of the journal. English is sound and the manuscript is well written. I was able to follow the analysis carried out by the authors even if I suggest some necessary modifications. One limit concerns the confusion on the used symbols: some of them are not clearly defined, both in the text and in the tables captions (e.g. H, D, t, x, y, ...), and some other are used to indicate more than one quantity (e.g., d). Moreover, the authors use acronyms before they are defined. I was wondering about the meaning of the term 'prediction interval' and if it is considered as an 'uncertainty band'. The comment of the previous reviewer and the reply of the authors shed light on this issue. I must say that, from my point of view, the term 'sensitivity analysis' would be more appropriate in this case. Minor comments: - explain what 'with a probability of occurrence larger than HQ5' means. - use always the past tense or the present tense throughout the manuscript.

**Dear Referee #2,**

**thank you very much for your comments.**

"One limit concerns the confusion on the used symbols: some of them are not clearly defined, both in the text and in the tables captions (e.g. H, D, t, x, y, ...), and some other are used to indicate more than one quantity (e.g., d). Moreover, the authors use acronyms before they are defined."

Thank you for pointing that out. We modified the manuscript accordingly.

*"I was wondering about the meaning of the term 'prediction interval' and if it is considered as an 'uncertainty band'. The comment of the previous reviewer and the reply of the authors shed light on this issue."*

It seems to me that the term "uncertainty band" refers to the same quantity as "prediction band" or "prediction interval". I would still stick to the latter to avoid further misunderstanding regarding the term "uncertainty", and as "prediction interval" seems to be more widely used according to Google. Nevertheless the definition given in Section 3 was improved.

*"Minor comments: - explain what 'with a probability of occurrence larger than HQ5' means. - use always the past tense or the present tense throughout the manuscript."*

We modified the manuscript accordingly.

**Modified Figure 1 (right)**

Added Figure 3